# Methodology for Optimizing Factors Affecting Road Accidents in Poland

**Piotr Gorzelanczyk * and Henryk Tylicki**

Transport Department, Stanislaw Staszic State University of Applied Sciences in Pila, ul. Podchorazych 10, 64-920 Pila, Poland
*   Correspondence: piotr.gorzelanczyk@ans.pila.pl

**Abstract:** With the rapid increase in the number of vehicles on the road, traffic accidents have become a rapidly growing threat, causing the loss of human life and economic assets. The reason for this is the rapid growth of the human population and the development of motorization. The main challenge in predicting and analyzing traffic accident data is the small size of the dataset that can be used for analysis in this regard. While traffic accidents cause, globally, millions of deaths and injuries each year, their density in time and space is low. The purpose of this article is to present a methodology for determining the role of factors influencing road accidents in Poland. For this purpose, multi-criteria optimization methods were used. The results obtained allow us to conclude that the proposed solution can be used to search for the best solution for the selection of factors affecting traffic accidents. Furthermore, based on the study, it can be concluded that the factors primarily influencing traffic accidents are weather conditions (fog, smoke, rainfall, snowfall, hail, or cloud cover), province (Lower Silesian, Lubelskie, Lodzkie, Malopolskie, Mazovian, Opolskie, Podkarpackie, Pomeranian, Silesian, Warmian-Masurian, and Greater Poland), and type of road (with two one-way carriageways; two-way, single carriageway road). Noteworthy is the fact that all days of the week also affect the number of vehicle accidents, although most of them occur on Fridays.

**Keywords:** multi-criteria optimization; traffic accident; factors affecting traffic accidents





## 1. Characteristics of the Issue

Road accidents are a serious social problem for any country. The causes of road accidents depend on various factors, such as weather conditions, the state of driver intoxication, car speed, etc. According to the World Health Organization [1], more than 1.35 million people die in road accidents each year, and millions more suffer serious injuries and long-term negative health consequences. Traffic accidents also generate economic losses. Year after year, the number of road accidents has been decreasing; the reason for this, in recent years, is primarily the COVID-19 pandemic. However, the number of vehicle accidents is very high (Figure 1), with an average of 62 automobile accidents occurring every day, in which six people are killed and 72 are injured. The above incidents result in increased medical costs, the need for repairs to vehicles and road infrastructure, and a negative impact on the environment (e.g., through fuel and operating fluid spills). For this reason, various measures are being taken to prevent traffic accidents and reduce their number. One such measure is the analysis of factors affecting the number of road accidents [2,3].

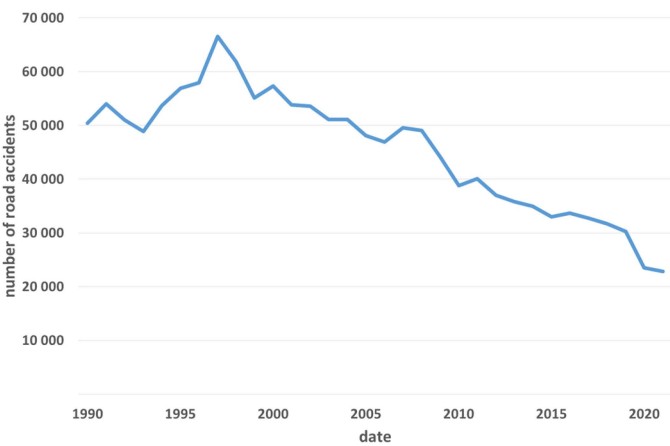

**Figure 1.** Number of road accidents in Poland between 1990 and 2021 [3].

Zhai et al. [4] and Holland et al. [5], in their studies, showed that pedestrians are at the highest risk of traffic accidents because they are less protected than motor vehicle passengers. In addition, they suffer more serious injuries than other road users. Their subsequent studies have shown that factors such as alcohol consumption, the age or gender of drivers, lighting, road conditions, pedestrian behavior, accident scene, vehicle, speed, and unfavorable weather affect the severity of pedestrian injuries. Adverse weather and inadequate lighting, especially of pedestrian crossings, often lead to more severe injuries in traffic accidents [6–8]. However, this depends on the area studied, e.g., a paper [9] showed that, in most cases, weather conditions have little effect on traffic accidents. A similar research topic can also be encountered in the work [10], in which the authors introduced a model of the probability of traffic accidents depending on driving time and current weather. The relationship between weather conditions and the number of traffic accidents has also been analyzed in other works [11–28]. In addition to weather conditions, traffic volume and driver behavior, such as their reaction time to the prevailing traffic situation, also have an impact on increasing the number of traffic accidents [14,29,30]. Eisenberg [16], in his work, studied the relationship between precipitation and traffic accidents in the US, where he showed that more traffic accidents occur during negative weather conditions. A similar topic was dealt with by Brodsky and Hakkert [31], who found that accidents increase by 100% during rainy conditions, while, in Denmark, the increase was negligible at around 10%. In contrast, Fridstrøm et al. [16] found that in Norway and Sweden, rainfall had no effect on the change in the number of traffic accidents. In Poland, on the other hand, the highest number of traffic accidents occurs during good weather conditions. Moreover, as the temperature increases and during good weather conditions, the number of traffic accidents increases [3,30,32].

Masello et al. [33], in their study, presented the influence of other factors on the number of traffic accidents; namely, they investigated the effect of driver assistance systems on improving road safety. The study was conducted in various traffic situations and weather conditions.

Similar studies have been conducted by other researchers. For example, Čubranić-Dobrodolac et al. [34] proposed an evaluation and decision support model for use when a driver should be tested for his tendency to be involved in traffic accidents, based on the estimation of the driver's psychological characteristics. In contrast, in another paper [35], the authors performed a study of the relationship between speed and drivers' ability to assess space in terms of their relationship with the occurrence of a driver's tendency to be involved in traffic accidents. An examination and understanding of the research trends in mining accidents and current scenarios related to this topic is described in another work [36].

Various methods of forecasting the number of accidents can be found in the literature. The most common methods of forecasting the number of road accidents are time series

methods [37,38], the disadvantages of which are the inability to assess the quality of the forecast based on outdated forecasts and frequent autocorrelation of the residual values of the component [39]. Procházka et al. [40] used a multiple seasonality model for forecasting, and Sunny et al. [41] used the Holt-Winters exponential smoothing method. Its limitations include the inability to introduce exogenous variables into the model [42,43].

To forecast the number of road accidents, the vector autoregression model was also used, the disadvantage of which is the need to have a large number of observations of variables in order to correctly estimate their parameters [44], as well as autoregression models by Monedero et al. for fatalities analysis [45] and Al-Madani using curve fit regression models [46]. These, in turn, require only simple linear relationships [47] and autoregression sequences (assuming that the series are already stationary) [48].

Atmospheric conditions also affect the relationship between three elements of traffic (the so-called safety triangle), namely the human being (and his psychomotor state, fatigue, stress, concentration), the vehicle (its technical condition, traffic speed, load), and the environment (road infrastructure) (Figure 2).

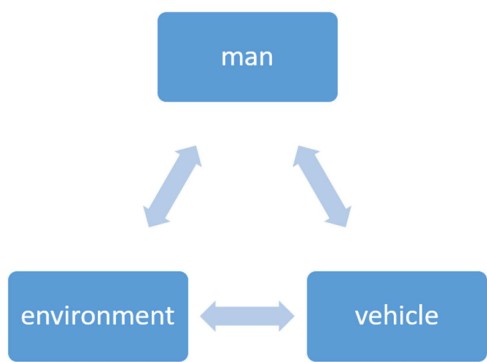

**Figure 2.** Safety triangle.

For the purpose of this work, definitions of terms related to the state of atmospheric conditions were defined [49]:

- Good atmospheric conditions are as follows:
  - air temperature > 3 deg;
  - no precipitation;
  - wind < 5.5 m/s;
  - visibility > 10 km;
  - daily differential pressure < 8 hPa.

- Bad weather conditions (if one of the following factors is met) are as follows:
  - slippery pavement (temperature < 3 °C and occurrence of precipitation);
  - driving rain (temperature > 0 °C, precipitation > 3 mm);
  - snowstorm (temperature < 0 °C, precipitation > 3 mm);
  - strong wind (wind > 10 ms/s);
  - dense fog (visibility < 300 m).

On the basis of the presented analysis of the literature related to the role of factors influencing the number of traffic accidents, it was found that some researchers of this issue assess their negative impact on the number of traffic accidents, while quite a large number state the absence of such an impact. Hence, there is a need to undertake the clarification of these discrepancies and related problems—for example, in terms of additional considerations related to the anthropotechnical system (driver, car, road) and the safety triangle (Figure 2) and the selection of factors affecting the number of road accidents in Poland. For this purpose, it is proposed to use multi-criteria optimization methods and tools, presented in the following sections of the paper. In addition, based on the above literature review, the authors did not find that the multi-criteria optimization method has been used in other

studies to predict the number of traffic accidents. For this reason, the authors will address this issue.

## 2. Multi-Criteria Optimization Model

When formulating an optimization task, it is difficult to specify a single scalar quality function *F*, since admissible solutions *X* may have many different properties whose values indicate the quality of the solution. Thus, it is necessary to formulate in this case an Optimization Task (ZO) with multiple (e.g., *N*) quality indicators in the form of a criterion function *F* [50–53]:

$$F: X \rightarrow R^N \tag{1}$$

This function assigns to each admissible solution $x \in X$ its numerical evaluation in the form of a vector:

$$F(x) = (F_1(x), \ldots, F_n(x), \ldots, F_N(x)) \in R^N \tag{2}$$

where

$N = \{1, \ldots, i, \ldots, n\}$—a set of quality indicator numbers;

$F_n(x)$—the value of the *n*-th quality indicator (*n*-th criterion function for the solution $x \in X$).

The formulation of the solution problem of determining the optimal solution is then presented as follows, where

- *A*—the space of solutions;
- *B*—the space of solution evaluations;
- F: A $\Rightarrow$ B—a criterion function, assigning to each solution $X \subset A$ its grade $Z \in B$ and assuming that the set of possible solutions A is not empty, a certain subset *X* (the set of acceptable solutions) can be selected, whereby

$$Z = F(X) = \{F(x) \in B \mid x \in X\} \tag{3}$$

After determining the set *X*, the mapping function *F*, and the dominance relation *Φ*, the optimization task (*ZO*) is formulated in the form

$$ZO = (X, F, \Phi) \tag{4}$$

where

$X = \{x_1, \ldots, x_n\}$—a set of possible solutions;

*F*—a criterion function for selecting possible solutions $F: X \Rightarrow R^N$

$$F(X) = (f_1(X), f_2(X), \ldots, f_n(x), \ldots, f_N(x)) \tag{5}$$

In the event that the *ZO* is considered for the $R^2$,

$$F(X) = (f_1(X), f_2(X)) \tag{6}$$

where the partial functions $f_1(X), f_2(X)$ can have a dominance relation preference structure, i.e., *Φ* MAX or MIN, respectively, where the dominant relationship *Φ* has a preference MAX:

$$\Phi = \{(c_1, c_2, \ldots, c_n, \ldots, c_N) \in C \times C\} \tag{7}$$

$$C = F(X) = \{(f_1(x), f_2(x)) \in R^2 : x \in X\} \tag{8}$$

where

*C*—image of the set *X* when mapped *F*;

$c_1, c_2$—points of space *C*;

or where the dominant relationship *Φ* has a preference MIN:

$$\Phi = \{(d_1, d_2, \ldots, d_n, \ldots, d_N) \in D \times D\} \tag{9}$$

$$D = F(X) = \{(f_1(x), f_2(x)) \in R^2 : x \in X\} \tag{10}$$

where

$D$—image of the set $X$ when mapped $F$;

$d_1, d_2$—points of space $D$.

Based on the above, a method for solving a multi-criteria optimization task is presented. Let the optimization task of determining possible solutions be

$$(X_1, F_1, \Phi_1) \tag{11}$$

where

$X_1$—the set of admissible solutions defined as

$$X_1 = \{x_{1,1}, x_{1,2}, x_{1,3}, x_{1,4}\} \tag{12}$$

$F_1$—quality indicator defined as $F_1 : X_1 \Rightarrow R^2$

$$F_1(X_1) = (f_{1,1}(x), f_{1,2}(x)) \tag{13}$$

$\Phi_1$—dominance relationship of preference, e.g., MAX, MAX.

To determine the set of dominant solutions $X_D^{\Phi 1}$ of the optimization task, find the product of the following sets $X_1^1$ and $X_1^2$:

$$X_1^1 = \{x^* \in X_1 : f_{1,1}(x^*) = \max f_{1,1}(x)\}; \, x \in X_1 \tag{14}$$

$$X_1^2 = \{x^* \in X_1 : f_{1,2}(x^*) = \max f_{1,2}(x)\}; \, x \in X_1 \tag{15}$$

where the quantities $f_{1,1}(x), f_{1,2}(x)$ are defined by appropriate relations, e.g.,

$$f_{1,1}(x) = ej(x) \text{ and } f_{1,2}(x) = kj(x) \tag{16}$$

Therefore, solve two tasks:

- maximize the function,

$$f_{1,1}(x) = e_j(x), \, x \in X_1; \, j = 1, \ldots, n \tag{17}$$

- maximize the function,

$$f_{1,2}(x) = k_j(x), \, x \in X_1; \, j = 1, \ldots, n \tag{18}$$

Then, determine the sets of $X_1^1$ and $X_1^2$,

$$X_1^1 = \{x \in^* X_1 : e_j(x^*) = \max e_j(x)\} \text{ for } x \in X_1 \tag{19}$$

$$X_1^2 = \{x^* \in X_1 : k_j(x^*) = \max k_j(x)\} \text{ for } x \in X_1 \tag{20}$$

and the set of dominant solutions as the product of the sets of $X_1^1$ and $X_1^2$,

$$X_D^{\Phi 1} = X_1^1 \cap X_1^2 \tag{21}$$

If the set $X_D^{\Phi 1}$ is empty, the set of non-dominated solutions $X_N^{\Phi 1}$ and the set of compromise solutions $X_K^{\Phi 1}$ are determined.

According to the remarks made above, the maximum value of the function (19) and the maximum value of the function (20) determine the coordinates of the ideal point $c^* = (c_1^*, c_2^*)$:

$$c_1^* = \max e_j(x); \, c_2^* = \max k_j(x)$$
$$x \in X_1; \, x \in X_1 \tag{22}$$

From the adopted form of the criterion function $F_1 = \{f_{1,1}, f_{1,2}\}$, it follows that for $c^*$ the maximum value of $e_j$ is demanded and the maximum value of $k_j$ is demanded.

In further considerations, the normalized index of the quality of the solution of the task (5,6) will be used, which is proposed to be

$$F_1^*(x) = \{f_{1,1}^*(x), f_{1,2}^*(x)\} \tag{23}$$

where

$$f_{1,1}^*(x) = \frac{f_{1,1}(x)}{c_1^{max}} \; ; \; f_{1,2}^*(x) = \frac{f_{1,2}(x)}{c_2^{max}} \tag{24}$$

whereby

$$c_1^{\,max} = \max_{x \,\in X_1} f_{1,1}(x), \; c_2^{\,max} = \max_{x \in X_1} f_{1,2}(x) \tag{25}$$

The advantage of this method of normalization is that the ratio is preserved after normalization. The highest value of the ratio is 1, and the lowest is greater than or equal to 0. The normalized ideal point then has the form

$$c^{**} = (c_1^{\,**}, c_2^{\,**}) \tag{26}$$

Due to the form of the set of admissible solutions $X_1$ (discreteness) to determine the set of its non-dominated solutions $X_N^{\Phi 1}$ and compromise solutions $X_K^{\Phi 1}$, a method is proposed to determine the approximate result (and therefore the solution) of the compromise for the norm $|\bullet|$, which is a measure of the distance of the results $c^* \in C^*$ from the ideal point $c^{**}$ [54,55].

Let $c^{**}$ denote the ideal point determined by relation (29) and $C^*$ the known set of normalized results:

$$C^* = \{c^{*i}\}, i = 1, \dots, n \tag{27}$$

where $c^{*i} = (c_1^{*i}, c_2^{*i})$, whereby

$$c_1^{*i} = \frac{c_1^i}{c_1^{max}} \; ; \; c_2^{*i} = \frac{c_2^i}{c_2^{max}} \tag{28}$$

In order to determine the compromise results, it is proposed to calculate the value of the standard $|\bullet|$ with the parameter $p = 2$:

$$r_i = \left| c_1^{**} - c_1^{*i} \right|^2 = \sqrt{\left( c_1^{**} - c_1^{*i} \right)^2 + \left( c_2^{**} - c_2^{*i} \right)^2} \tag{29}$$

and selecting such a result $c^o$, which would minimize the calculated values of $r_i$ norms, e.g., $x_1^{\,o} = x_{1,3}$

$$x_1^{\,o} = c^o = \min r_i \tag{30}$$

An interpretation of the above method is shown in Figure 3.

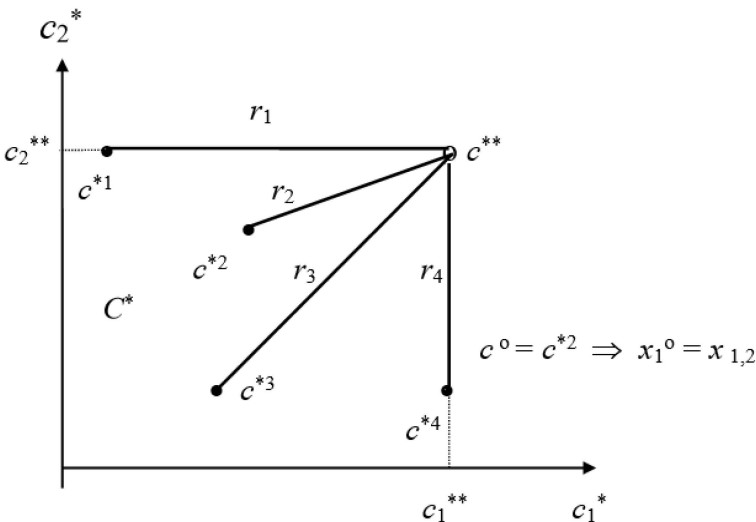

**Figure 3.** Graphical interpretation of the solution to the optimization task [56].

### 3. Optimization of Factors Affecting the Number of Traffic Accidents

In the case of the proposed multi-criteria optimization methodology for factors affecting the number of traffic accidents, many solutions to this problem are possible. One way of solving this problem is presented in the solution scheme of the optimization task of determining the optimal set of factors affecting road accidents in Poland (Figure 4). For this reason, the set of admissible solutions $X$ can be [54,55].

$$X = \{X_1\} \tag{31}$$

where

$X_1 = \{x_{1,1}, \dots , x_{1,n}\}$—is a set of atmospheric factors affecting the number of traffic accidents,

$$X_1 = (x_{1,1}, x_{1,2}, \dots , x_{1,35}) \tag{32}$$

where

atmospheric conditions:

- $x_{1,1}$—good weather;
- $x_{1,2}$—fog, smoke;
- $x_{1,3}$—rainfall;
- $x_{1,4}$—snowfall, hail;
- $x_{1,5}$—blinding sun;
- $x_{1,6}$—cloudy;
- $x_{1,7}$—strong wind;

day of the week:

- $x_{1,8}$—Monday;
- $x_{1,9}$—Tuesday;
- $x_{1,10}$—Wednesday;
- $x_{1,11}$—Thursday;
- $x_{1,12}$—Friday;
- $x_{1,13}$—Saturday;
- $x_{1,14}$—Sunday;

province:

- $x_{1,15}$—Lower Silesia;
- $x_{1,16}$—Kujawsko-pomorskie;
- $x_{1,17}$—Lubelskie;
- $x_{1,18}$—Lubuskie;

- $x_{1,19}$—Lodzkie;
- $x_{1,20}$—Lesser Poland;
- $x_{1,21}$—Mazovian;
- $x_{1,22}$—Opolskie;
- $x_{1,23}$—Subcarpathian;
- $x_{1,24}$—Podlaskie;
- $x_{1,25}$—Pomeranian;
- $x_{1,26}$—Silesian;
- $x_{1,27}$—Swietokrzyskie;
- $x_{1,28}$—Warmian-Masurian;
- $x_{1,29}$—Greater Poland;
- $x_{1,30}$—Zachodniopomorskie.

  road type:

- $x_{1,31}$—highway;
- $x_{1,32}$—expressway;
- $x_{1,33}$ –with two one-way roadways;
- $x_{1,34}$—road—one-way;
- $x_{1,35}$—two-way, single carriageway.

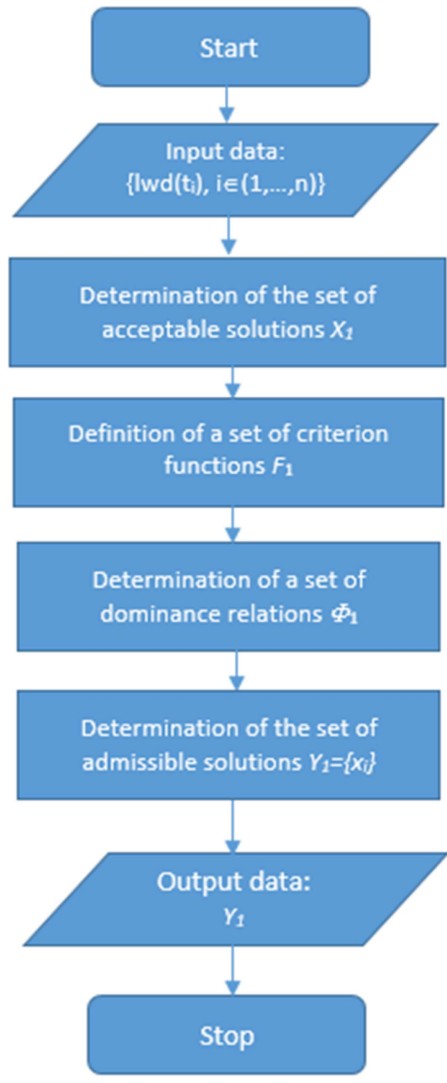

**Figure 4.** Multi-criteria optimization scheme for determining the optimal set of factors influencing road accidents in Poland.

If we have a set of $F_1$, we can define the vector solution quality index $F$ as

$$F_1 = F_1(X_1) = (f_{1,1}(X_1), f_{1,2}(X_1), f_{1,3}(X_1)) \tag{33}$$

and determine $F_1$ criterion functions, for optimizing weather conditions affecting the number of traffic accidents in Poland, for example, as [54,55]:

$$F_1 = (f_{1,1}, f_{1,2}, f_{1,3}) \tag{34}$$

where

$f_{1,1}$—method of maximum relative change in factor affecting *lwd*;
$f_{1,2}$—method of maximum change in gradient of factor affecting *lwd*;
$f_{1,3}$—method of maximum change in factor affecting *lwd*;

$$f_{1,1} = \frac{1}{K} \sum_{i=1}^{K} \frac{lwd(t_i)}{lwd_{max}} \tag{35}$$

$$f_{1,2} = \frac{1}{K-1} \sum_{i=1}^{K} |lwd(t_{i+1}) - lwd(t_i)| \tag{36}$$

$$f_{1,3} = \frac{1}{K} \sum_{i=1}^{K} |lwd(t_i) - lwd_{max}| \tag{37}$$

where

$n$—number of measurements;
$lwd_{max}$—maximum *lwd* value of analyzed measurements;
$lwd(t_i)$—number of traffic accidents over time $t_i$;
$lwd(t_{i+1})$—number of traffic accidents over time $t_{i+1}$.
$\Phi_1$—criteria dominance relationship vector indicator of solution quality $F$ ([51,57]:

$$\Phi_1 = \{\Phi_{1,1}, \Phi_{1,2}, \Phi_{1,3}\} \tag{38}$$

where

$\Phi_{1,1}$—dominance relationship in terms of $f_{1,1}$ (MAX);
$\Phi_{1,2}$—dominance relationship in terms of $f_{1,2}$ (MAX);
$\Phi_{1,3}$—dominance relationship in terms of $f_{1,3}$ (MAX).
Then, the solution to the optimization task of determining the optimal set of factors influencing road accidents in Poland *ZO* takes the form

$$ZO = < X_1, F_1, \Phi_1 > \tag{39}$$

This is then implemented according to the following algorithm.

1.  Normalization of criterion space—space $D^*$

Set of normalized results $D^*$.

$$D^* = \{d^{*i}\}, i = 1, \ldots, n; d^{*i} = (d_1^{*i}, d_2^{*i}, d_3^{*i}) \tag{40}$$

2.  Determination of the coordinates of the ideal point—$d^{**}$.

$$d^{**} = (d_1^{**}, d_2^{**}, d_3^{**})$$
$$d_1^{**} = \max f^*_{1,1}(x), d_2^{**} = \max f^*_{1,2}(x)$$
$$d_3^{**} = \max f^*_{1,3}(x), x \in X_1 \tag{41}$$

3.  Calculation of the value of the standard with parameter $p = 2$—$r_j(D^*)$.

Norm $|\bullet|$ is a measure of the distance of $d^* \in D^*$ results from the ideal point $d^{**}$.

$$r_j(D^*) = |d^{**} - d^{*i}| = \sqrt{(d_1^{**} - d_1^{*i})^2 + (d_2^{**} - d_2^{*i})^2 + (d_3^{**} - d_3^{*i})^2} \tag{42}$$

Determination of the optimal result $x_1{}^o$ in an optimization task—for example, if $x_1{}^o = x_{1,4}$,

$$x_1{}^o = d^o = \min r_i; \text{ because } d^o = \min r_3 \tag{43}$$

Then, the optimal solution, i.e., to determine the largest role of the factor influencing road accidents in Poland $x_i \in X_1$, for example, is the factor $x_{1,4}$ (the optimal set of one-element solutions is obtained—one factor).

If there is a need to obtain an optimal set of multi-element solutions, which may be the case here, we use the procedure for determining the set of values $\{r_i\}$ [57] and, based on them, determine a multi-element set of solutions (the optimal set of factors significantly affecting the number of traffic accidents in Poland).

## 4. Example of Optimization of Factors Affecting the Number of Road Accidents in Poland

In order to solve the task of multi-criteria optimization, a computer program, "Multi-Criteria Optimization Task 2017", was developed (work of selection of the means of transportation), which allows [57]:

- the presentation of a set $X_j$ and selection of elements $x_i X_j$;
- the presentation of the set $F_j$ and selection, by the computer program operator, of the elements $f_i F_j$ and the dominance relation $_{i j}$;
- data entry according to two options (option 1—manual data entry ($f_i F_j$ values), option 2—calculation of $f_i F_j$ values) based on data obtained during experimental or simulation studies;
- visualization of the solution of the optimization task (calculation and reporting of calculation results—Tables 1–3).

**Table 1.** $F_1$ criteria values and dominance relationship $\Phi_1$.

|  | $x_{1,1}$ | $x_{1,2}$ | $x_{1,3}$ | $x_{1,4}$ | $x_{1,5}$ | $x_{1,6}$ | $x_{1,7}$ | $x_{1,8}$ | $x_{1,9}$ | $x_{1,10}$ | $x_{1,11}$ | $x_{1,12}$ |
|---|---|---|---|---|---|---|---|---|---|---|---|---|
| $f_{1,1}$ | 0.73 | 0.53 | 0.70 | 0.50 | 0.78 | 0.68 | 0.53 | 0.73 | 0.74 | 0.74 | 0.74 | 0.75 |
| $f_{1,2}$ | 1622.45 | 132.20 | 700.40 | 539.85 | 69.15 | 1541.80 | 62.50 | 305.60 | 297.50 | 323.70 | 261.55 | 338.45 |
| $f_{1,3}$ | 9721.62 | 0.53 | 2041.29 | 1188.33 | 178.81 | 3600.67 | 269.00 | 2162.38 | 1911.57 | 1993.81 | 1981.33 | 2168.48 |

|  | $x_{1,13}$ | $x_{1,14}$ | $x_{1,15}$ | $x_{1,16}$ | $x_{1,17}$ | $x_{1,18}$ | $x_{1,19}$ | $x_{1,20}$ | $x_{1,21}$ | $x_{1,22}$ | $x_{1,23}$ | $x_{1,24}$ |
|---|---|---|---|---|---|---|---|---|---|---|---|---|
| $f_{1,1}$ | 0.70 | 0.72 | 0.74 | 0.59 | 0.60 | 0.78 | 0.85 | 0.75 | 0.61 | 0.23 | 0.77 | 0.43 |
| $f_{1,2}$ | 397.00 | 277.75 | 229.80 | 96.50 | 146.25 | 40.65 | 224.60 | 406.60 | 431.40 | 247.20 | 148.80 | 96.75 |
| $f_{1,3}$ | 2492.81 | 1935.95 | 914.71 | 1100.14 | 1205.62 | 223.14 | 726.57 | 1267.81 | 3399.33 | 3942.71 | 543.86 | 1301.67 |

|  | $x_{1,25}$ | $x_{1,26}$ | $x_{1,27}$ | $x_{1,28}$ | $x_{1,29}$ | $x_{1,30}$ | $x_{1,31}$ | $x_{1,32}$ | $x_{1,33}$ | $x_{1,34}$ | $x_{1,35}$ | $\Phi_1$ |
|---|---|---|---|---|---|---|---|---|---|---|---|---|
| $f_{1,1}$ | 0.76 | 0.69 | 0.71 | 0.73 | 0.68 | 0.73 | 0.64 | 0.49 | 0.76 | 0.83 | 0.73 | **MAX** |
| $f_{1,2}$ | 184.05 | 297.05 | 81.20 | 140.05 | 392.75 | 103.20 | 40.70 | 23.60 | 294.25 | 81.40 | 1732.15 | **MAX** |
| $f_{1,3}$ | 883.10 | 2189.29 | 631.62 | 599.95 | 1709.76 | 598.05 | 171.05 | 229.05 | 1620.90 | 253.00 | 12,116.14 | **MAX** |

**Table 2.** Visualization of the results of solving the optimization task.

| F/X | $f_{1,1}$ | MAX $(f_{1,1})$ | $f_{1,1}*$ | $f_{1,1}**$ | $f_{1,2}$ | MAX $(f_{1,2})$ | $f_{1,2}*$ | $f_{1,2}**$ | $f_{1,3}$ | MAX $(f_{1,3})$ | $f_{1,3}*$ | $f_{1,3}**$ |
|---|---|---|---|---|---|---|---|---|---|---|---|---|
| $x_{1,1}$ | 0.73 | 0.85 | 1.17 | 3.66 | 1622.4 | 1732.15 | 1.07 | 73.4 | 9721.62 | 12,116.14 | 0.8 | 1 |
| $x_{1,2}$ | 0.53 | | 1.61 | | 132.2 | | 13.1 | | 0.53 | | 0 | |
| $x_{1,3}$ | 0.7 | | 1.21 | | 700.4 | | 2.47 | | 2041.29 | | 0.17 | |
| $x_{1,4}$ | 0.5 | | 1.7 | | 539.85 | | 3.21 | | 1188.33 | | 0.1 | |
| $x_{1,5}$ | 0.78 | | 1.09 | | 69.15 | | 25.05 | | 178.81 | | 0.01 | |
| $x_{1,6}$ | 0.68 | | 1.25 | | 1541.8 | | 1.12 | | 3600.67 | | 0.3 | |
| $x_{1,7}$ | 0.53 | | 1.59 | | 62.5 | | 27.71 | | 269 | | 0.02 | |
| $x_{1,8}$ | 0.73 | | 1.16 | | 305.6 | | 5.67 | | 2162.38 | | 0.18 | |
| $x_{1,9}$ | 0.74 | | 1.14 | | 297.5 | | 5.82 | | 1911.57 | | 0.16 | |
| $x_{1,10}$ | 0.74 | | 1.15 | | 323.7 | | 5.35 | | 1993.81 | | 0.16 | |
| $x_{1,11}$ | 0.74 | | 1.15 | | 261.55 | | 6.62 | | 1981.33 | | 0.16 | |
| $x_{1,12}$ | 0.75 | | 1.13 | | 338.45 | | 5.12 | | 2168.48 | | 0.18 | |
| $x_{1,13}$ | 0.7 | | 1.22 | | 397 | | 4.36 | | 2492.81 | | 0.21 | |
| $x_{1,14}$ | 0.72 | | 1.19 | | 277.75 | | 6.24 | | 1935.95 | | 0.16 | |
| $x_{1,15}$ | 0.74 | | 1.14 | | 229.8 | | 7.54 | | 914.71 | | 0.08 | |
| $x_{1,16}$ | 0.59 | | 1.45 | | 96.5 | | 17.95 | | 1100.14 | | 0.09 | |
| $x_{1,17}$ | 0.6 | | 1.41 | | 146.25 | | 11.84 | | 1205.62 | | 0.1 | |
| $x_{1,18}$ | 0.78 | | 1.1 | | 40.65 | | 42.61 | | 223.14 | | 0.02 | |
| $x_{1,19}$ | 0.85 | | 1 | | 224.6 | | 7.71 | | 726.57 | | 0.06 | |
| $x_{1,20}$ | 0.75 | | 1.14 | | 406.6 | | 4.26 | | 1267.81 | | 0.1 | |
| $x_{1,21}$ | 0.61 | | 1.39 | | 431.4 | | 4.02 | | 3399.33 | | 0.28 | |
| $x_{1,22}$ | 0.23 | | 3.66 | | 247.2 | | 7.01 | | 3942.71 | | 0.33 | |
| $x_{1,23}$ | 0.77 | | 1.1 | | 148.8 | | 11.64 | | 543.86 | | 0.04 | |
| $x_{1,24}$ | 0.43 | | 1.97 | | 96.75 | | 17.9 | | 1301.67 | | 0.11 | |
| $x_{1,25}$ | 0.76 | | 1.12 | | 184.05 | | 9.41 | | 883.1 | | 0.07 | |
| $x_{1,26}$ | 0.69 | | 1.23 | | 297.05 | | 5.83 | | 2189.29 | | 0.18 | |
| $x_{1,27}$ | 0.71 | | 1.19 | | 81.2 | | 21.33 | | 631.62 | | 0.05 | |
| $x_{1,28}$ | 0.73 | | 1.16 | | 140.05 | | 12.37 | | 599.95 | | 0.05 | |
| $x_{1,29}$ | 0.68 | | 1.25 | | 392.75 | | 4.41 | | 1709.76 | | 0.14 | |
| $x_{1,30}$ | 0.73 | | 1.17 | | 103.2 | | 16.78 | | 598.05 | | 0.05 | |
| $x_{1,31}$ | 0.64 | | 1.32 | | 40.7 | | 42.56 | | 171.05 | | 0.01 | |
| $x_{1,32}$ | 0.49 | | 1.75 | | 23.6 | | 73.4 | | 229.05 | | 0.02 | |
| $x_{1,33}$ | 0.76 | | 1.11 | | 294.25 | | 5.89 | | 1620.9 | | 0.13 | |
| $x_{1,34}$ | 0.83 | | 1.03 | | 81.4 | | 21.28 | | 253 | | 0.02 | |
| $x_{1,35}$ | 0.73 | | 1.16 | | 1732.15 | | 1 | | 12116,14 | | 1 | |

**Table 3.** Distance values $r_j$ $(x_i)$.

| | $x_{1,1}$ | $x_{1,2}$ | $x_{1,3}$ | $x_{1,4}$ | $x_{1,5}$ | $x_{1,6}$ | $x_{1,7}$ | $x_{1,8}$ | $x_{1,9}$ | $x_{1,10}$ | $x_{1,11}$ | $x_{1,12}$ |
|---|---|---|---|---|---|---|---|---|---|---|---|---|
| $r_j$ | 72.37 | **13.20** | **2.76** | **3.63** | 25.07 | **1.70** | 27.76 | **5.79** | **5.94** | **5.48** | **6.72** | **5.24** |
| | $x_{1,13}$ | $x_{1,14}$ | $x_{1,15}$ | $x_{1,16}$ | $x_{1,17}$ | $x_{1,18}$ | $x_{1,19}$ | $x_{1,20}$ | $x_{1,21}$ | $x_{1,22}$ | $x_{1,23}$ | $x_{1,24}$ |
| $r_j$ | **4.53** | **6.35** | **7.62** | 18.01 | **11.93** | 42.63 | **7.78** | **4.41** | **4.26** | **7.91** | **11.69** | 18.01 |
| | $x_{1,25}$ | $x_{1,26}$ | $x_{1,27}$ | $x_{1,28}$ | $x_{1,29}$ | $x_{1,30}$ | $x_{1,31}$ | $x_{1,32}$ | $x_{1,33}$ | $x_{1,34}$ | $x_{1,35}$ | |
| $r_j$ | **9.48** | **5.96** | 21.37 | **12.42** | **4.59** | 16.83 | 42.58 | 73.42 | **5.99** | 21.30 | **1.83** | |

In view of the necessity of obtaining the optimal set of multi-element solutions, the average value $r_{av}$ determined from the set of values $\{r_j(x_i), j = 1, \dots , 35\}$ was determined, which is the criterion for classifying the admissible solutions $x_i \in X_j$ into the set of optimal solutions $x_i^o \in X_j^o$ according to the principle expressed by the relation

$$x_i^o \in X_j^o = r_j <= r_{av} \qquad (44)$$

If the determined value of $r_{av}$ = 15.33, then the elements of the set of solutions of optimal factors significantly affecting the number of traffic accidents $X_j^o$, according to the above classification criterion, respectively, are as follows:
weather conditions:

- fog, smoke;
- rainfall;
- snowfall or hail;
- cloud cover;

  days of the week:

- Monday;
- Tuesday;
- Wednesday;
- Thursday;
- Friday;
- Saturday;
- Sunday;

  province:

- Lower Silesian;
- Lubelskie;
- Lodzkie;
- Małopolskie;
- Mazovian;
- Opolskie;
- Podkarpackie;
- Pomeranian;
- Silesian;
- Warmian-Masurian;
- Greater Poland;

  type of road:

- with two one-way carriageways;
- a two-way, single carriageway road.

## 5. Summary

The methodology presented above for the use of multi-criteria optimization procedures using a multi-criteria optimization model and some elements of the $ZO$ optimization task (partial criteria of the criterion function $F_1$) and elements of the dominance relation $_1$) allows us to conclude that it can be used to optimize factors affecting the number of traffic accidents in Poland. The main advantage of the presented algorithm is its versatility; it follows that it will probably be possible to apply the procedures of the presented methodology in situations where the elements of the criterion function will be quantitative and qualitative, and when there is a need to obtain a multi-element or single-element optimal set of solutions.

In addition, based on the study, it can be concluded that the factors mainly affecting traffic accidents are weather conditions (fog, smoke; rainfall; snowfall or hail; cloud cover), province (Lower Silesian; Lubelskie; Lodzkie; Małopolskie; Mazovian; Opolskie; Podkarpackie; Pomeranian; Silesian; Warmian-Masurian; Greater Poland), and type of road (with two one-way carriageways; two-way, single carriageway road). Noteworthy is the fact that all days of the week also affect the number of traffic accidents, although most accidents occur on Fridays.

In further research, the authors plan to optimize methods for forecasting the number of road accidents in Poland and to optimize methods for forecasting the number of road accidents depending on the synergy of factors affecting the number of accidents.

**Author Contributions:** P.G.: conceptualization; data curation; formal analysis; funding acquisition; investigation; project administration; resources; software; supervision; validation; visualization; writing—original draft; writing—review and editing; H.T.: methodology. All authors have read and agreed to the published version of the manuscript.

**Funding:** The article was financed by the university's own funds.

**Institutional Review Board Statement:** Not applicable.

**Informed Consent Statement:** Not applicable.

**Data Availability Statement:** The research was conducted using public data.

**Conflicts of Interest:** The authors declare that they have no known competing financial interests or personal relationships that could have appeared to influence the work reported in this paper.

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
