# Peer review of "Methodology for Optimizing Factors Affecting Road Accidents in Poland"

_forecasting, doi:10.3390/forecast5010018_

Round 1
Reviewer 1 Report
Dear authors,
I recommend adding a comparison of your results to other published approaches, for example:
Čubranić-Dobrodolac M, Švadlenka L, Čičević S, Trifunović A, Dobrodolac M. Using the Interval Type-2 Fuzzy Inference Systems to Compare the Impact of Speed and Space Perception on the Occurrence of Road Traffic Accidents. Mathematics. 2020; 8(9):1548. https://doi.org/10.3390/math8091548
Čubranić-Dobrodolac, M., Švadlenka, L., Čičević, S., Dobrodolac, M. (2020). Modeling driver propensity for traffic accidents: a comparison of multiple regression analysis and fuzzy approach. International Journal of Injury Control and Safety Promotion. 27(2), 156–167. https://doi.org/10.1080/17457300.2019.1690002
Ismail, S. N., Ramli, A., & Aziz, H. A. (2021). Research trends in mining accidents study: A systematic literature review. Safety science, 143, 105438.
Ferreira-Vanegas, C. M., Vélez, J. I., & García-Llinás, G. A. (2022). Analytical methods and determinants of frequency and severity of road accidents: a 20-year systematic literature review. Journal of advanced transportation, 2022.
Author Response
Dear Professor,
Thank you for your valuable comments on my article. In accordance with your suggestion, I have added other approaches in the paper. The changes have been marked in yellow.
Regards
Piotr Gorzelańczyk

Reviewer 2 Report
I have gone through the paper “Methodology for optimizing factors affecting road accidents in Poland,” and I have the following observations:
- Equation number (10) is not in the appropriate place.
- Equation number (42) is not written in the proper paragraph.
- The discussion section should be incorporated.
- Results have not been analyzed; classified case-to-case situations should do it.
The paper should be a major revision with the above-given suggestions.
Author Response
Dear Professor,
Thank you for your valuable comments on my article. As per your suggestion, I have added the following in the paper:
- I changed the location of equation 10
- I left equation 42 in the same place because I believe it is in the right place. This is due to the algorithm of the optimization process to determine the distance rj from the ideal point
- I modified the discussion and results section,
The changes have been highlighted in yellow.
Greetings
Piotr Gorzelańczyk
Reviewer 3 Report
See Word document for suggestions

Author Response
Dear Professor,
Thank you for your valuable comments on my article. In accordance with your suggestion, I have added the following in the paper:
- I corrected the abstract
- I modified formulas 7, 8, 10, 16to reduce ambiguity
- I corrected figure 3
- I modified the discussion and results section,
- I added other approaches in this area.
- I modified the discussion and results section,
The changes have been highlighted in yellow.
Greetings
Piotr Gorzelańczyk

Round 2
Reviewer 2 Report
Done as suggested, Recommended for publication.
Author Response
Dear Professor
Thank you for your help.
Best regards
Piotr Gorzelanczyk

Reviewer 3 Report
1. Increase the horizontal comparison between the model in the text and other models, and explain the advantages of the model in the text.
2. The four and five parts of the article from data processing to conclusions are abrupt, and the whole article is about multi-criteria optimization. Except for the traffic factors included in the variables, other parts do not reflect the content related to traffic accidents. The traffic language should be used for correlation analysis after data processing, explain the application of this method in the factors affecting traffic accidents, and strengthen the correlation between research methods and research contents.
Author Response
Professor, thank you for your valuable comments. The following things have been corrected in the article:
- Issues related to days of the week have been corrected.
- Other methods of forecasting the number of traffic accidents have been characterized
- Contributions of co-authors have been corrected
Best Regards
Piotr Gorzelanczyk